# Efficient Implementation of a Robot-Assisted Radical Cystectomy Program in a Naïve Centre Experienced in Open Radical Cystectomy and Other Robot-Assisted Surgeries: A Comparative Analysis of Perioperative Outcomes and Complications

**DOI:** 10.3390/cancers17152532

**Published:** 2025-07-31

**Authors:** Gianluca Giannarini, Gioacchino De Giorgi, Maria Abbinante, Carmine Franzese, Jeanlou Collavino, Fabio Traunero, Marco Buttazzi, Antonio Amodeo, Angelo Porreca, Alessandro Crestani

**Affiliations:** 1Urology Unit, Santa Maria della Misericordia University Hospital, Piazzale Santa Maria della Misericordia 15, 33100 Udine, Italy; gioacchino.degiorgi@asufc.sanita.fvg.it (G.D.G.); maria.abbinante@asufc.sanita.fvg.it (M.A.); carmine.franzese@asufc.sanita.fvg.it (C.F.); jeanlou.collavino@asufc.sanita.fvg.it (J.C.); fabio.traunero@asufc.sanita.fvg.it (F.T.); alessandro.crestani@uniud.it (A.C.); 2Department of Medicine, University of Udine, 33100 Udine, Italy; marcobuttazzi98@gmail.com; 3Oncological Urology Unit, Veneto Institute of Oncology IOV—IRCCS, 35128 Padua, Italy; antonio.amodeo@iov.veneto.it; 4Urology Unit, Humanitas Gavazzeni, 24125 Bergamo, Italy; angelo.porreca@hunimed.eu

**Keywords:** bladder cancer, radical cystectomy, urinary diversion, ileal conduit, cutaneous ureterostomy, robot-assisted surgery, minimally invasive surgery, complications, morbidity, learning curve, training, education

## Abstract

This study examined whether a hospital that had never performed robotic bladder removal surgery before could safely and effectively start doing so. The hospital already had experience with traditional open bladder removal and other types of robotic procedures. The researchers wanted to find out if switching to robotic surgery would improve patient recovery without increasing risks. They compared results from patients who had robotic surgery to those who had the traditional open approach. They found that robotic surgery led to fewer complications, less blood loss, and shorter hospital stays, even though it was a new procedure for the hospital team. These findings suggest that with proper preparation and support, hospitals can safely adopt robotic techniques, potentially improving outcomes for bladder cancer patients. This research may encourage other hospitals to transition to robotic surgery, expanding access to modern, less invasive, and less morbid treatments.

## 1. Introduction

Both robot-assisted radical cystectomy (RARC) and open radical cystectomy (ORC) are currently recommended by the European Association of Urology (EAU) guidelines as valid options for the treatment of nonmetastatic muscle-invasive bladder cancer, as well as very high-risk and Bacillus Calmette–Guérin (BCG)-refractory, BCG-relapsing, or BCG-unresponsive high-risk non-muscle-invasive bladder cancer [1].

RARC is a technically demanding intervention that involves several complex surgical steps, including bowel handling for urinary diversion (UD). Its adoption has been slower and narrower compared with other major oncological robot-assisted surgeries (RAS), such as radical prostatectomy and radical or partial nephrectomy, primarily because of its steep learning curve. Several centres, even those experienced in RAS, remain hesitant to establish and maintain a RARC program, likely because of perceived organisational challenges, prolonged operating room (OR) times, and the risk of unsustainably inferior perioperative outcomes and higher complications during the early stages of transition. These concerns are particularly relevant to the UD phase of the procedure, especially when performed fully intracorporeally, as it significantly contributes to procedural morbidity—potentially even more so than the extirpative phase itself.

Earlier evidence from meta-analysis of nonrandomised and randomised studies demonstrated that RARC was associated with generally superior perioperative outcomes—primarily reduced blood loss and transfusion rate, and, to a lesser extent, shorter length of stay (LOS)—than ORC, but morbidity, quality of life and oncological outcomes were comparable [2,3]. Notably, these studies included patients who underwent extracorporeal UD. More recent evidence from clinical trials suggests that the use of RARC with intracorporeal UD could provide even greater benefits than those observed after RARC with extracorporeal UD or ORC, namely fewer overall complications and better quality of life in the early postoperative period, albeit at the cost of longer OR times and higher rate of ureteric strictures [4,5,6].

However, in the absence of robust real-world comparative evidence, there currently remains uncertainty regarding the relative effectiveness of RARC with intracorporeal UD versus ORC. Given the challenges posed by the complexity of the intervention, a crucial point is the transition phase from ORC to RARC, especially if intracorporeal UD is performed. To our knowledge, there is no standardised pathway, and only scarce data are available from analyses of the outcomes of such transition.

The aims of this study were (1) to assess the feasibility and outcomes of the implementation of a RARC program in a naïve centre experienced in ORC and other RAS and (2) to compare the perioperative and postoperative outcomes of RARC versus ORC cases treated at our centre before the start of the RARC program.

## 2. Materials and Methods

### 2.1. Study Design and Patients

The Santa Maria della Misericordia University Hospital in Udine is a tertiary referral academic medical centre serving a large area of the Friuli-Venezia Giulia region (approximately 1.2 million inhabitants) in Northeast Italy. The Urology Unit is a high-volume centre for uro-oncological interventions. A RAS program has been running since June 2017 but included only radical prostatectomy and radical or partial nephrectomy as well as upper urinary tract reconstructive surgery. RC was performed only with an open approach, with a total of 400 procedures in the latest decade.

The RARC program started in June 2023, when the new chairman (A.C.), an experienced open, laparoscopic, and robotic surgeon with a previous track record of 60 RARC cases with intracorporeal UD performed at the Oncological Urology Unit, Veneto Institute of Oncology IOV—IRCCS, Padua, Italy, joined the team. The study period was chosen to encompass the implementation phase from ORC to RARC, with a similar number of patients for each approach. Fifty consecutive bladder cancer patients undergoing elective RARC up to January 2025 were included in this prospective analysis, while the latest 50 consecutive bladder cancer patients operated on with elective ORC from September 2020 served as controls. Patients scheduled for, and receiving, orthotopic neobladder (n = 1); those in whom cystectomy was aborted because of intraoperatively detected unresectable disease (n = 3); and those undergoing an emergency cystectomy due to intractable bleeding or bladder perforation (n = 8) were excluded.

This study was conducted in accordance with good clinical practice guidelines and the principles of the Declaration of Helsinki. All patients provided a written informed consent and authorised data collection for scientific purposes with anonymous publication. The institutional review board waived the approval of this study, since all patients received a standard-of-care treatment.

### 2.2. Surgical Technique and Proctoring Program

ORC procedures were performed as previously described [7,8]. RARC procedures were performed using the da Vinci^®^ Xi platform and included fully intracorporeal UD. The surgical technique was as previously described [9,10]. Briefly, the intervention was divided into the following steps. The cystectomy phase consisted of (1) isolation and division of the ureters, (2) incision of the Douglas pouch and preparation of the space between the prostate and rectum, (3) preparation of the lateral space and division of the vesicoprostatic pedicles, (4) preparation of the anterior space and division of the Santorini complex and urethra, and (5) bilateral pelvic lymph node dissection, where appropriate. The UD phase consisted of (1) retrosigmoid transposition of the left ureter, (2) harvesting of the bowel segment and ileoileal anastomosis, (3) ureteroileal anastomosis, (4) stoma creation, in the case of ileal conduit, or creation of a cutaneous ureterostomy. Ileal conduit was performed in both groups with the Wallace technique.

Two dedicated RARC-naïve surgeons with high experience in open pelvic surgery and with prior experience in other RAS (radical prostatectomy and radical/partial nephrectomy, >30 cases each) were proctored. No dedicated nursing staff was selected. Both naïve surgeons and nursing staff were instructed with prerecorded video material on the surgical steps in the two weeks preceding the first case (theoretical training). No preclinical training was performed. The clinical modular training consisted of three phases. In the first phase, the proctored surgeons table-assisted the proctoring surgeon for the entire procedure (5 cases each). In the second phase, cystectomy and, if applicable, bilateral pelvic lymph node dissection were performed by the proctored surgeons (10 cases each), while UD was always performed by the proctoring surgeon. In the third phase, cystectomy and, if applicable, bilateral pelvic lymph node dissection were always performed by the proctoring surgeon, while UD was performed by the proctored surgeons (10 cases each).

### 2.3. Postoperative Management

An enhanced recovery protocol was followed for all patients, which included no preoperative dietary restrictions, no bowel preparation, opioid-sparing general anaesthesia with restrictive fluid administration, no nasogastric tube, no preplanned postoperative stay in the intensive/intermediate care unit, standardised postoperative pain control, no total parenteral nutrition, early nutritionist-guided oral diet, and early mobilisation, as previously reported [11].

Pelvic drains were removed between postoperative day (POD) 2 and 4. Ureteric catheters were removed on POD 7 and 8 in the ORC cases and on POD 10 and 11 in the RARC ones. Thromboembolic prophylaxis included compression stockings until full mobilisation and low molecular weight heparin up to 4 postoperative weeks.

Criteria for hospital discharge included: passage of stool, tolerance to regular solid diet, satisfactory pain control with oral agents alone, no fever, drains removed, good competence with urinary stoma, home healthcare planned if needed.

The first follow-up visit was planned 3 months after surgery and included a physical examination and serum exams, including electrolytes, creatinine, liver function tests, and an abdomen/pelvis ultrasound examination. For our analysis, further follow-up visits were not considered.

### 2.4. Data and Study Outcomes

Data were extracted from a prospectively maintained database where data were collected by medical staff via patient charts or in-person or telephone interviews. Preoperative parameters were: age at surgery, sex, body mass index, Charlson comorbidity index, American Society of Anesthesiologists score, pre-existing upper urinary tract dilatation, clinical stage according to the 2010 TNM staging system, and receipt of neoadjuvant chemotherapy. Perioperative parameters were: OR time, estimated blood loss (EBL), and intraoperative complications graded according to the EAU intraoperative adverse incident classification [12], and LOS. Pathological parameters were: extension of primary tumour and lymph node involvement according to the 2010 TNM system, tumour grading, lymph node yield, and status of soft tissue surgical margins. Ninety-day postoperative complications were defined and scored using the Clavien–Dindo classification [13], and grouped by severity and type according to the Memorial Sloan Kettering Cancer Center complication grading system [14]. Ninety-day readmissions were also recorded. Quality criteria for accurate and comprehensive reporting of surgical outcomes were followed as recommended by the EAU guidelines [15] (Appendix A).

Study outcomes were: (1) completion rate of a fully intracorporeal RARC procedure and (2) perioperative outcomes and postoperative complications in the RARC versus ORC cohort.

### 2.5. Statistical Analysis

Parametric continuous variables were reported as mean ± standard deviation, whereas median and interquartile range (IQR) was used for nonparametric continuous variables. The Mann–Whitney U-test and Pearson’s χ^2^ test were used to compare continuous nonparametric and categorical variables, as appropriate. Finally, we explored whether some clinical characteristics, including type of approach (open vs. robotic), were associated with postoperative complications using multivariable logistic regression models. All clinical records were inserted into a dedicated database and analysed using SPSS v. 24.0 (IBM Corp., Armonk, NY, USA) software. All reported *p* values are two-sided, and statistical significance was set at *p* < 0.05.

## 3. Results

Patient characteristics are detailed in Table 1. The two groups were comparable for all variables, except for a higher receipt of neoadjuvant chemotherapy in the RARC cohort (*p* = 0.03). Notably, median age was 77 (IQR 73–84) yrs and 76 (IQR 72–80) yrs in the RARC and ORC cohort, respectively (*p* = 0.88). Clinical stage was T3/4 in 7/50 (14%) and 8/50 (16%) cases in the RARC and ORC cohort, respectively (*p* = 0.55). The rate of ileal conduit diversion was 27/50 (54%) and 31/50 (62%) in the RARC and ORC cohort, respectively (*p* = 0.11).

In the RARC cohort, the procedure was completed fully intracorporeally in all patients. Perioperative outcomes are reported in Table 2. Notably, EBL, rate of perioperative transfusions, and LOS were all significantly lower or shorter in the RARC versus ORC cohort. OR time was significantly shorter in the RARC cohort receiving an ileal conduit. No intraoperative complications were observed in either group.

No patients were lost to follow-up. Within 90 days from surgery, 37 patients (74%) in the RARC group and 49 patients (98%) in the ORC group experienced at least one postoperative complication (*p* < 0.01), with most patients (27 [54%] and 41 [82%], respectively, *p* < 0.01) experiencing multiple complications. Details of the worst single postoperative complications are reported in Table 3. Notably, grade 3 or higher complications were significantly less frequent in the RARC than the ORC cohort (11 [11%] versus 21 [21%], *p* = 0.045).

Overall, 107 and 172 postoperative complications were observed in 37 and 49 patients of the RARC and ORC cohort, respectively (*p* = 0.03). Among these complications, 20 and 31 events were severe (*p* = 0.19). The most common complications in both cohorts were infectious followed by gastrointestinal (Appendix A).

No difference in readmission rate was found in the two cohorts (Table 2). Median time to readmission for ORC and RARC patients was 31 days (IQR 24–65) and 36 days (IQR 30–46), respectively (*p* = 0.57).

Finally, 14 patients (14%) died. Among these, 12 patients (12%)—5 and 7 in the RARC and ORC cohort, respectively—died within 2 months from surgery because of complications, and 2 patients (2%) died between 2 and 3 months after surgery because of cancer progression.

On exploratory multivariable logistic regression, the robot-assisted approach was the only independent predictor of a lower risk of overall complications (Table 4).

## 4. Discussion

We showed that a RARC program can be safely and rapidly implemented in a RARC-naïve unit with surgeons experienced in ORC and other RAS with no need for a dedicated, structured training curriculum. Moreover, our findings corroborate previous studies indicating that RARC is associated with improved perioperative outcomes and a lower incidence of complications compared with ORC.

Several findings of our study are noteworthy. First, the implementation of a RARC program was safe and relatively rapid. Our proctoring program did not follow any standardised training, although it was based on the known concept of clinical modular training [16]. To our knowledge, no RARC curriculum has been formally validated. Only a proposal for a curriculum for RARC with ileal conduit in male patients has been published [17]. Moreover, there are only limited data on the analysis of the outcomes of the transition from ORC to RARC. Miller et al. assessed the outcomes of the implementation phase of RARC into an existing ERAS pathway for ORC in a designated regional cancer centre in the UK [18]. A cohort of 114 patients treated over a 20-month period was examined. After an initial period with extracorporeal UD, the fully intracorporeal procedure was adopted. Only one case was converted to ORC. Compared with historical institutional ORC cases, RARC was associated with only a marginal gain in LOS and 30-day mortality, but with a higher rate of 30-day major complications and readmissions. These results may have been due to the double learning curve the surgeons had to go through, the inclusion of neobladder cases, and/or the positive effect of a long-consolidated ERAS pathway [19]. Another UK study assessed the perioperative outcomes of the transition from extracorporeal to intracorporeal UD during RARC in a centre with a long expertise in RARC [20]. Over a 30-month period encompassing the transition phase, 68 patients received extracorporeal, and 59, intracorporeal, ileal conduits. In general, no significant differences were observed in perioperative outcomes between the two groups, confirming the safety of the transition. OR time for ileal conduit creation, EBL, and rate of overall 30-day complications were even reduced in the intracorporeal cohort.

Second, we included consecutive patients to reflect real-world practice with no case selection. The older median age of our cohort reflects the shifting demographics of bladder cancer, highlighting the growing challenge of managing this disease in an aging population. In a recent comparative study, RARC with ICUD in well-selected elderly patients (aged ≥80 years) achieved a tolerable high-grade complication rate. The 90-day postoperative mortality rate was driven by cancer progression, and the non-cancer-related rate was equivalent to that of patients aged <80 years [21]. Notably, a large proportion of our unselected octogenarian patients had also multiple severe comorbidities, which may explain the relatively high mortality rate.

Third, we observed significant benefits of RARC with intracorporeal UD compared with ORC, which are particularly noteworthy given the elderly population treated at our centre. Our findings contribute to the ongoing debate by providing new real-world evidence. Similarly, a recent and larger real-world comparative study demonstrated improved perioperative outcomes, a higher lymph node count, better ureteroileal stricture outcomes, and comparable cancer-specific and overall survival rates after RARC with intracorporeal UD versus ORC, with a median follow-up of 42 months [22].

Fourth, we followed the state-of-the-art methodology for reporting surgical complications. Implementing a prospective system for meticulously collecting intra- and postoperative complications is essential to providing valuable insights that can enhance patient management for RARC with intracorporeal UD as well as other complex procedures, while identifying key areas for future research. Several studies have demonstrated that only through the adoption of such comprehensive systems of depiction of complication-intervention events and grading can the morbidity associated with this complex surgical procedure be accurately assessed and compared across centres and techniques [23,24,25,26,27,28,29]. This is also crucial when ensuring appropriate postoperative follow-up beyond the classical short-term period. Notably, a recent study with an extended observation time of at least one year after RARC with intracorporeal UD revealed that up to 28% of severe complications occurred beyond the traditional 90-day postoperative window [30].

Our study is not devoid of limitations. First, the proctored surgeons had vast experience in open surgery, including ORC, and also had prior experience in RAS. Whether prior experience in ORC and other RAS might have played a role in the safe and rapid implementation of a fully intracorporeal RARC programme remains unknown. Our results might not be generalisable to other proctoring programs where surgeons are naïve to open or robotic exposure. Second, the favourable outcomes observed during the implementation phase may be attributed to the substantial active involvement of the proctoring surgeon. It remains to be determined whether these outcomes can be sustained in the subsequent phase of the transition process (adoption phase), when proctored surgeons begin performing procedures independently. Third, neobladder cases were omitted from proctoring, because it was felt that the learning process for upfront RARC with intracorporeal neobladder would have been more challenging and, rather, this would have taken profit from a prior proficiency in less demanding types of UD. Fourth, no oncological or quality-of-life data were measured. Fifth, our findings need to be interpreted within the inherent limitations of a nonrandomised observational study design with selection bias and residual confounding. Furthermore, because of the relatively limited sample size, caution should be exerted when interpreting certain subgroup analyses.

## 5. Conclusions

We showed that a RARC program can be safely implemented in a RARC-naïve unit with surgeons experienced in other RAS with no need for a dedicated and structured training curriculum, nor for case-selection. While we focused on the initial implementation phase of the transition, we also showed benefits in terms of perioperative outcomes and complications with RARC and intracorporeal UD compared with ORC in our elderly and multimorbid population. Our findings may encourage other centres to transition to robotic surgery for their bladder cancer patients, thereby expanding access to less invasive and less morbid treatment options.

## Figures and Tables

**Table 1 cancers-17-02532-t001:** Demographic, preoperative, surgical, and pathological characteristics of the two cohorts of radical cystectomy patients included in the comparative analysis.

Variables	Total Cases (n = 100)	RARC Group (n = 50)	ORC Group (n = 50)	*p* Value
Age, years, median (IQR)	76 (72–83)	77 (73–84)	76 (72–80)	0.88
Male gender, n (%)	75 (75)	35 (70)	40 (80)	0.57
BMI, kg/m^2^, median (IQR)	25.88 (23.04–27.97)	25.76 (22.87–27.39)	25.98 (23.63–28.40)	0.43
Charlson comorbidity index, n (%)				0.23
- 0	41 (41)	23 (46)	18 (36)	
- 1	19 (19)	9 (18)	10 (20)	
- 2	14 (14)	6 (12)	8 (16)	
- >2	26 (26)	12 (24)	14 (28)	
ASA score, n (%)				0.36
- 2	44 (44)	24 (48)	20 (40)	
- 3	52 (52)	24 (48)	28 (56)	
- 4	4 (4)	2 (4)	2 (4)	
TUR-BT, n (%)				0.99
- complete	16 (16)	8 (16)	8 (16)	
- incomplete	84 (84)	42 (84)	42 (84)	
Clinical tumour stage, n (%)				0.54
- T1	12 (12)	6 (12)	6 (12)	
- Tis	9 (9)	5 (10)	4 (8)	
- T2	64 (64)	32 (64)	32 (64)	
- T3	10 (10)	6 (12)	4 (8)	
- T4	5 (5)	1 (2)	4 (8)	
Clinical node stage, n (%)				0.82
- N0	85 (85)	43 (86)	42 (84)	
- N1	10 (10)	6 (12)	4 (8)	
- N2	5 (5)	1 (2)	4 (8)	
Upper urinary tract dilatation, n (%)				0.45
- absent	75 (75)	42 (84)	33 (66)	
- unilateral	14 (14)	5 (10)	9 (18)	
- bilateral	11 (11)	3 (6)	8 (16)	
Receipt of neoadjuvant chemotherapy, n (%)	21 (19)	14 (28)	7 (14)	0.03
Urinary diversion, n (%)				0.21
- Ileal conduit	59 (59)	27 (54)	31 (62)	
- Cutaneous ureterostomy	41 (41)	23 (46)	19 (38)	
Pathological tumour stage, n (%)				0.23
- T0	11 (11)	4 (8)	7 (14)	
- Tis	3 (3)	2 (4)	1 (2)	
- T1	14 (14)	10 (20)	4 (8)	
- T2	21 (21)	12 (24)	9 (18)	
- T3	32 (32)	10 (20)	22 (44)	
- T4	17 (17)	10 (20)	7 (14)	
Pathological node stage, n (%)				0.15
- Nx	29 (29)	19 (38)	10 (20)	
- N0	53 (53)	23 (46)	30 (60)	
- N1	8 (8)	4 (8)	4 (8)	
- N2	8 (8)	2 (4)	6 (12)	
Positive soft tissue margins, n (%)	6 (6)	3 (6)	3 (6)	0.99

ASA: American Society of Anesthesiologists; BMI: body mass index; IQR: interquartile range; ORC: open radical cystectomy; RARC: robot-assisted radical cystectomy; TUR-BT: transurethral resection of bladder tumour.

**Table 2 cancers-17-02532-t002:** Perioperative outcomes of the two radical cystectomy cohorts included in the comparative analysis.

Variables	Total Cases (n = 100)	RARC Group (n = 50)	ORC Group (n = 50)	*p* Value
Operating room time (ileal conduit), min, median (IQR)	287 (250–310)	255 (245–298)	300 (284–312)	<0.01
Operating room time (cutaneous ureterostomy), min, median (IQR)	200 (164–240)	182 (173–242)	215 (151–238)	0.11
Estimated blood loss, mL, median (IQR)	200 (90–500)	100 (0–275)	425 (200–600)	0.01
Perioperative blood transfusions, n (%)	35 (35%)	12 (24%)	23 (46%)	0.02
Length of stay, days, median (IQR)	9 (7–14)	7 (6–9)	12 (9–20)	0.01
Ninety-day readmissions, n (%)	27 (27%)	15 (30%)	12 (24%)	0.22

IQR: interquartile range; ORC: open radical cystectomy; RARC: robot-assisted radical cystectomy.

**Table 3 cancers-17-02532-t003:** Worst single 90-day postoperative complications scored according to Clavien–Dindo classification in the two cohorts of radical cystectomy patients included in the comparative analysis.

Grade	Overall Complications, n (%)	RARC Group (N = 50), n (%)		ORC Group (N = 50), n (%)		Complication Type	Treatment
1	34 (34)	17 (34)	10	17 (34)	6	Fever of unknown origin	Antipyretics, antimicrobials
			4		1	Urinary tract infection	Antimicrobials
			-		1	Bedsore	Ointment
			-		2	Ileus	Nasogastric tube
			-		2	Constipation	Enema
			1		1	Supraventricular tachycardia	Beta-blocker
			1		3	Nausea/vomit	Antiemetics
			1		1	Diarrhoea	Loperamide
2	20 (20)	9 (18)	8	11 (22)	9	Anaemia	Blood transfusions
			-		1	Atrial fibrillation	Pharmacological cardioversion
			1		1	Agitation/delirium	Haloperidol
3a	9 (9)	3 (6)	1	6 (12)	1	Ureteric catheter dislocation	Ureteric catheter replacement
			-		3	Ureteric catheter obstruction	Ureteric catheter replacement
			2		-	Ureteric obstruction	Ureteric catheter insertion
			-		2	Wound dehiscence	Surgical synthesis
3b	7 (7)	3 (6)	2	4 (8)	2	Adhesion-related small bowel obstruction	Laparotomy with lysis
			-		1	Rectum perforation	Terminal sigma colostomy
			1		-	Femur fracture	Osteosynthesis
			-		1	Wound dehiscence	Surgical synthesis
4a	3 (3)	0 (0)	-	3 (6)	1	Pulmonary embolism	Thrombolysis
			-		1	Acute kidney injury	Dialysis
			-		1	Myocardial infarction	Percutaneous transluminal coronary angioplasty
4b	1 (1)	0 (0)	-	1 (2)	1	Septic shock	Intensive cardiopulmonary support, dialysis
5	12 (12)	5 (10)	2	7 (14)	4	Septic shock	Intensive cardiopulmonary support, dialysis
			-		1	Haemorrhagic shock	Laparotomy, hypogastric artery embolisation
			-		1	Non-ST-elevation myocardial infarction	Intensive cardiopulmonary support
			-		1	Acute heart failure	Intensive cardiopulmonary support
			1		-	Epileptic seizure with coma	Intensive neurological and cardiopulmonary support
			2		-	Acute kidney injury	Intensive cardiopulmonary support, dialysis

ORC: open radical cystectomy; RARC: robot-assisted radical cystectomy.

**Table 4 cancers-17-02532-t004:** Multivariable logistic regression analysis exploring predictors of overall postoperative complications as per Table 4 in the entire cohort of radical cystectomy patients.

Variables	OR	95% CI	*p* Value
Age (years)	1.03	0.97, 1.08	0.07
Male (vs. female)	1.11	0.94, 1.85	0.51
ASA score ≥ 3 (vs. 2)	1.12	0.89, 2.01	0.22
BMI (kg/m^2^)	1.02	0.97, 1.07	0.11
Robot-assisted (vs. open) approach	0.81	0.65, 0.95	0.01

ASA: American Society of Anesthesiologists; BMI: body mass index; CI: confidence interval; OR: odds ratio.

## Data Availability

The data that support the findings of this study are available from the corresponding author upon reasonable request.

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
