# Peer review of "Efficient Implementation of a Robot-Assisted Radical Cystectomy Program in a Naïve Centre Experienced in Open Radical Cystectomy and Other Robot-Assisted Surgeries: A Comparative Analysis of Perioperative Outcomes and Complications"

_cancers, 2025, doi:10.3390/cancers17152532_

Round 1
Reviewer 1 Report
Comments and Suggestions for Authors
This is a study concerning the comparison of Robotic and Open Cystectomy in a center with no previous experience in RARC. The paper is well written with sound conclusions. Nevertheless I have some remarks.
1)It is important to stress the fact that surgeons had enough experience to this kind of operations (even though it was open) and also most important is the fact that they had also robotic experience. We know that except from the UD all the other steps are not so difficult to perform if someone has experience in robotic prostatectomy or upper tract surgery. In my point of view these factors somewhat diminish the strength of this study and authors must expand on this subject
2)Table 4 has too much information that I think it is not needed for a paper that studies surgical procedures. Please revise it in groups and in severity and this must be enough
3)From the proctoring strategy we see that most of the cases (or many of its steps) were performed by the proctor. 10 out of 50 were performed entirely from the proctor, 20 the proctor performed the UD which is the step from which the most complication derive from and the next 20 the proctor performed the cystectomy part. I feel that the outcomes are flawed from this strategy. I would like to see the results of the procedures that these surgeons produced when they performed the cases all alone after the proctorship program. This could reflect the real data and support the title of the study.
Author Response
This is a study concerning the comparison of Robotic and Open Cystectomy in a center with no previous experience in RARC. The paper is well written with sound conclusions. Nevertheless I have some remarks.
1) It is important to stress the fact that surgeons had enough experience to this kind of operations (even though it was open) and also most important is the fact that they had also robotic experience. We know that except from the UD all the other steps are not so difficult to perform if someone has experience in robotic prostatectomy or upper tract surgery. In my point of view these factors somewhat diminish the strength of this study and authors must expand on this subject.
We fully respect the Reviewer’s view on this point, although this assumption has never been demonstrated in a prospective ad hoc study. For this reason, rather than acknowledging a diminished strength and scientific value of our findings, we have been more cautious, and stated in the Discussion in the original manuscript that “… the proctored surgeons had a vast experience in open surgery, including ORC, and also had prior experience in RAS. Whether prior experience in ORC and other RAS might have played a role in the safe and rapid implementation of a fully intracorporeal RARC programme remains unknown. Our results might not be generalisable to other proctoring programs where surgeons are naïve to open or robotic exposure”.
2) Table 4 has too much information that I think it is not needed for a paper that studies surgical procedures. Please revise it in groups and in severity and this must be enough.
While we acknowledge that Table 4 contains a lot of information, this is exactly in line with the state-of-the-art methodology for reporting complications after surgery. As stated in the Methods, and commented also in the Discussion with expanded data, “Quality criteria for accurate and comprehensive reporting of surgical outcomes were followed as recommended by the EAU guidelines [15] (Supplementary Table 1)”. The same methodology (with the same table) has been adopted in other studies (ref #21 in our manuscript, or doi: 10.1016/j.eururo.2019.08.011, which we have now added to the reference list). For these reasons, we have opted to keep this table as is, but we have now moved it to the supplementary material.
3) From the proctoring strategy we see that most of the cases (or many of its steps) were performed by the proctor. 10 out of 50 were performed entirely from the proctor, 20 the proctor performed the UD which is the step from which the most complication derive from and the next 20 the proctor performed the cystectomy part. I feel that the outcomes are flawed from this strategy. I would like to see the results of the procedures that these surgeons produced when they performed the cases all alone after the proctorship program. This could reflect the real data and support the title of the study.
We fully agree with this comment. This is the next step of the transition process from ORC to RARC after implementation, i.e. early adoption, and we strongly believe it deserves a separate publication. We have already started working on that, but we would require to collect a cohort of adequate size, which does inevitably take time. For the time being, we have tried to further emphasize this limitation in the Discussion. We have also clarified the wording used throughout the text: we are referring to “transition” as the entire process, and to “implementation” and “adoption” as the two subsequent steps.
Reviewer 2 Report
Comments and Suggestions for Authors
The paper under review is a retrospective study evaluating the implementation of robot assisted radical cystectomy in a centre with no previous experience in this procedure (but with experience in open radical cystectomy). Such trials are important and necessary because they help province the necessary insight for adoption of new minimally invasive techniques in different centers.
Introduction is clear, it is not very long and provides a perfect insertion in the subject for the reader.
Materials and methods are is exhaustive. The complete methodology of the study is explained in detail. The surgery is explained in a comprehensive manner.
Results chapter is very well made. Every detail is comprehensively explained and tables are used to help the reader understand them better. All complications of the surgeries are presented together with the ways they were treated.
Discussions are in line with the subject of the paper.
Conclusions are concise and provide a clear insight on the problem disputed in the study.
For the reasons stated I propose for publication.
Author Response
The paper under review is a retrospective study evaluating the implementation of robot assisted radical cystectomy in a centre with no previous experience in this procedure (but with experience in open radical cystectomy). Such trials are important and necessary because they help province the necessary insight for adoption of new minimally invasive techniques in different centers.
Introduction is clear, it is not very long and provides a perfect insertion in the subject for the reader.
Materials and methods are is exhaustive. The complete methodology of the study is explained in detail. The surgery is explained in a comprehensive manner.
Results chapter is very well made. Every detail is comprehensively explained and tables are used to help the reader understand them better. All complications of the surgeries are presented together with the ways they were treated.
Discussions are in line with the subject of the paper.
Conclusions are concise and provide a clear insight on the problem disputed in the study.
For the reasons stated I propose for publication.
We thank the Reviewer for the positive comments.
Reviewer 3 Report
Comments and Suggestions for Authors
Dear authors, after reading your paper, I think it has some drawbacks that need to be addressed.
Your study has a non-randomised observational design, which inherently has limitations such as selection bias and residual confounding, also did not measure oncological or quality of life data.
The proctored doctors possessed considerable expertise in open surgery, specifically Open Radical Cystectomy, as well as experience in various robot-assisted procedures. The influence of this prior experience on the efficient and swift execution of the robotic program remains ambiguous. Consequently, the findings may not apply to other proctoring programs in which surgeons possess no prior expertise in open or robotic procedures.
Author Response
Dear authors, after reading your paper, I think it has some drawbacks that need to be addressed.
Your study has a non-randomised observational design, which inherently has limitations such as selection bias and residual confounding, also did not measure oncological or quality of life data.
We fully agree with this comment. These limitations had already been addressed in the Discussion in the original manuscript (“… no oncological or quality of life data were measured … our findings need to be interpreted within the inherent limitations of a non-randomised observational study design with selection bias and residual confounding”).
The proctored doctors possessed considerable expertise in open surgery, specifically Open Radical Cystectomy, as well as experience in various robot-assisted procedures. The influence of this prior experience on the efficient and swift execution of the robotic program remains ambiguous. Consequently, the findings may not apply to other proctoring programs in which surgeons possess no prior expertise in open or robotic procedures.
We fully agree with this comment. This limitation had already been addressed in the Discussion in the original manuscript (“First, the proctored surgeons had a vast experience in open surgery, including ORC, and also had prior experience in RAS. Whether prior experience in ORC and other RAS might have played a role in the safe and rapid implementation of a fully intracorporeal RARC programme remains unknown. Our results might not be generalisable to other proctoring programs where surgeons are naïve to open or robotic exposure”).